# Depth-Resolved Elemental Analysis on Moving Electrode Foils with Laser-Induced Breakdown Spectroscopy

**DOI:** 10.3390/s23031082

**Published:** 2023-01-17

**Authors:** Carl Basler, Moritz Kappeler, Daniel Carl

**Affiliations:** Fraunhofer Institute for Physical Measurement Techniques IPM, 79110 Freiburg, Germany

**Keywords:** laser-induced breakdown spectroscopy (LIBS), inline measurement, lithium-ion battery, depth profiling

## Abstract

In this study, a new method for the inline measurement of depth profiles on a continuously moving sample with laser-induced breakdown spectroscopy is presented. The ablation profile is generated by ablating the sample with a burst of laser pulses, where the emission spectrum of each laser-induced plasma is analyzed on a spectrometer. A Q-switched Nd:YAG laser at 1064 nm with 10 mJ pulse energy, 6 ns pulse duration and 100 Hz repetition rate was used. The focusing lens for the pulsed laser and a deflection mirror are mounted on a moving stage, which is precisely aligned in height and orientation to the movement of a conveyor belt transporting the sample. The stage speed is actively synchronized to the speed of the moving sample by a wheel encoder to assure that all laser pulses hit the same position at the sample. The feasibility for depth-resolved elemental analysis on moving samples is shown for coatings of electrode foils for lithium-ion batteries. The coating homogeneity was measured at a speed up to 17 m/min. For a 100 μm coating, 10 laser pulses were needed to measure a full depth profile.

## 1. Introduction

Laser-induced breakdown spectroscopy (LIBS) is often used for the elemental analysis of solid and homogeneous materials, such as alloy determination [1] in the metal industry and rock composition [2] in the mining industry, as it is fast, all-optical and does not need extensive sample preparation in a laboratory. However, LIBS can also be used for depth-resolved elemental analysis. Therefore, the material is ablated with multiple laser pulses at the same spot with the plasma emission of each ablation pulse being analyzed. This technique is used to measure the composition in depth for some applications like paintings [3], metals [4] and electrode foils [5].

Depth profiling can also be used to measure the thickness of a coating. For a known ablation rate, the thickness can be derived from the signal of coating and base material with pulse number. Dwivedi et al. [6] showed that depth profiles are in good agreement with SIMS profiles. They used a Nd:YAG laser with 5 ns pulse duration to ablate a 5–15 μm coating of Be, O, C, H and D on a tungsten substrate. The resolution and precision of the ablation process can be improved by using an ultra-short laser pulse (pico- or femtosecond), as shown by [7] and in our group [8,9]. In all these fields, LIBS competes with the more common techniques like SIMS, ICP-OES/MS and REM-EDX.

In contrast to LIBS, all other mentioned techniques are not suitable for inline measurements on a conveyor belt in a production process.

Although widely used for coating thickness analysis, X-ray fluorescence (XRF) is also conflicting with fast-moving samples since the acquisition time is typically in the range of seconds up to minutes. In contrast to LIBS, the technique is furthermore insensitive to elements of low atomic number such as Na and lower, and the ionizing radiation may pose potential harm to the health of the operator [10,11].

In order to play to the strength of LIBS, Balzer et al. showed how to realize the measurement of coating thickness with a depth profile for galvanized steel on a moving sample [12]. They used several laser bursts with different pulse energy, each irradiated on a laterally displaced steel sheet position. The ablation depth is on the order of the coating thickness. In this way, the thickness of a zinc coating on a moving galvanized steel sheet can be measured. Another approach using multiple pulses with well-controlled energy and separated by short measurement intervals on a similar but stationary sample was demonstrated in [13]. To our knowledge, there is no publication so far in which the depth-resolved analysis of the homogeneity of the elemental distribution in motion with LIBS has been performed.

This measurement could be of great interest for the production of electrode foils for lithium-ion batteries to monitor the homogeneity of its constituents. The feasibility of LIBS for the 2D and 3D measurement of the elemental distribution in electrode foils has been shown in several publications [5,14,15,16,17,18]. In all publications, the used samples were not in movement.

In this study, we demonstrate a new method to measure the depth-resolved elemental distribution of Li-Ion battery cathodes with time-integrated LIBS at a speed of up to 17 m/min. We calibrate the measurement with homogeneous samples with defined composition to quantify the distribution and compare the measurement at different speeds of movement. The concentration was determined from the ratio between the spectral lines of carbon and the Nickel–Mangan–Cobalt complex.

## 2. Materials and Methods

Lithium-ion batteries consist of an anode, a cathode, an electrolyte and a separator. The anode is based on a thin copper foil with a homogeneous coating of graphite mixed with a binder (e.g., polyvinylidene fluoride (PVDF)). The cathode is based on a thin aluminum foil with a coating of active material made of mixed oxide beads, e.g., Lithium–Nickel–Mangan–Cobalt oxide (NMC), with 1–10 μm diameter embedded in a graphite and binder matrix. The coating is made by solving the materials in a solvent such as water or acetone, stirring the solution to get a homogeneous slurry and then applying the emulsion as a thin film with a doctor’s blade. Afterwards, the solvent is evaporated in a drying process. On a microscale, this coating is inherently not as homogeneous as the anode, because the NMC beads are randomly distributed. However, it is important for stability and performance of the battery cell that the constituents NMC, binder and graphite are homogeneously distributed within the thickness of the layer [19]. By comparing electron microprobe analysis of cathode cross-sections with LIBS measurements, we showed that fluctuations in the normalized LIBS spectrum are dominated by the inhomogeneity of the dispersion [20]. The relevant parameter for the homogeneity to be measured is the 3D distribution of the carbon–NMC ratio.

For this study, we used anodes and cathodes from the production line from VARTA microbatteries. The coating of the cathodes contains 94% (weight) NMC beads (ratio Ni:Mg:Co 6:2:2), 1% binder and 5% graphite. For the calibration, we further used samples with 5, 6, 7 and 8% binder. The coating thickness is about 90 μm for anodes and 60 μm for cathodes.

A sketch of the experimental setup used for the coating measurement with LIBS is depicted in Figure 1. The laser used is a nanosecond Nd:YAG laser (Quantum Light Instruments, Q2-100) with 1064 nm wavelength and 6 ns pulse duration. For our experiments, the pulse energy was set to 10 mJ; the repetition rate was 100 Hz. The beam was focused with a lens with f = 200 mm to a spot with a waist size of approximately 80 μm. For metals, we expect an ultra-short pulsed laser to be much more suited for depth profiling as the material melts with a nanosecond pulse, different layers of the material can mix, and the ablation is less well defined [9]. In the case of electrode foils, melting is not crucial as the metals are embedded in a carbon matrix which vaporizes quickly. The focusing lens and a deflection mirror are mounted on a stage (OWIS prototype with linear motor, 15 cm travel) together with the plasma detection optics. The alignment of the optical axis of the laser beam and the stage is performed with a camera monitoring the shift of the laser focus while moving the stage. A laser triangulation distance sensor (optoNCDT 1320 by micro-epsilon) monitors the distance between the optics and the sample. The distance between the focusing lens and the conveyor belt is aligned with the triangulation sensor signal. Furthermore, the direction of movement of the conveyor belt and the stage are precisely aligned. In this way, the focus of the laser can follow the movement of the electrode foil. The conveyor belt has an adjustable speed of 0.1–17 m/min. To compensate for changes in the speed of the conveyor belt and to compensate for small variations, an encoder wheel (SICK DFS60A-S4PC65536) measures the speed continuously. The encoder has a wheel with 200 mm perimeter and 65.536 counts per revolution corresponding to a resolution of 3 μm. The belt speed is directly coupled to the encoder signal. A Czerny–Turner spectrometer with a CCD-Line (Avantes Avaspec ULS2048L-EVO, 187–263 nm) is used to detect the spectral information of the plasma. It has a 10 μm entrance slit and a spectral resolution of 0.1 nm. The used integration time is 1.05 ms, which starts before the laser pulse and integrates over the whole plasma lifetime. The plasma light is imaged into a fiber by two lenses with f = 50 mm.

## 3. Results

### 3.1. Calibrating Measurements on Stationary Samples

The experiments were performed on anodes and cathodes for lithium ion-batteries. In order to keep the measurement time short, the laser pulse energy was tuned to ablate the coating in approximately 10 pulses, which was the case at 10 mJ. Figure 2 shows an image of the crater on the cathode after laser ablation. The image was taken with a 3D laser scanning microscope.

Figure 3 shows the crater depth on the cathode as a function of the pulse number. The ablation rate is 5 μm per pulse; the coating has a thickness of 60 μm.

The plasma emission was evaluated in the range between 190 nm and 235 nm. One atomic spectral line of carbon is at 193.1 nm. Between 215 nm and 235 nm are numerous lines of Nickel, Cobalt and Manganese (see Figure 4). The lines were assigned by comparison to the NIST Atomic spectra database [21].

Four samples with different carbon concentrations (5–8%) were used to calibrate the system for a quantitative evaluation of the carbon content. Figure 5 shows the linear dependency of the ratio of the line intensity between carbon line and metal lines. The ratio is evaluated as
(1)$$ratio=\frac{I_{C}-I_{CBack}}{(\sum_{i}^{N}I_{Met_{i}}-I_{MetBack_{i}})/N}$$
where $I_{C}$ is the intensity of the carbon line, $I_{CBack}$ is a background value near the carbon line, $I_{Met_{i}}$ is one of N metal lines and $I_{Met_{Backi}}$ is a corresponding background value.

As the spectra of single pulses show a big variance, from pulse-to-pulse spectra from 400 positions with 10 spectra at each position (depth profile) were averaged. Pulse-to-pulse variations are mainly due to the distribution of the NMC beads [20].

A decrease in carbon signal has been observed systematically for subsequent laser pulses at the same spot (Figure 6). We attribute this behavior to the ablation rate of carbon, which is higher compared to NMC beads. This assumption is supported by microscope images showing uncovered NMC beads after the first two laser pulses. Consequently, we assume that the initial few laser pulses lead to over-average ablation of carbon with respect to NMC. The first laser pulse thus appears unsuitable for quantitative depth-profiling of carbon concentration. The second pulse still shows higher carbon content in the plasma signal, but the signal quickly approaches a constant value. This curve is used to calibrate the concentration of carbon with depth by dividing the course of the depth profile by the black course in Figure 6.

### 3.2. Measurements on Moving Samples

A depth profile on a moving sample requires a precise alignment of the measurement system to the conveyor belt in three dimensions. The crater has a diameter of 150 μm; 10 pulses were used to measure a depth profile, and all 10 pulses must hit the same crater. To prove the alignment, we measure the number of pulses needed to ablate the full coating. Therefore, we use an anode with an inherent homogeneous graphite coating and evaluate the first appearance of copper in the spectrum. Figure 7 shows the spectrum of the eighth and the 13th laser pulse with carbon lines at 193 and 248 nm and copper lines in between. For a slow-moving sample, eight–nine pulses are needed; up to 12 m/min, this value does not change significantly.

Figure 8 shows a simulation of the number of pulses needed to ablate the coating with a linear shift applied from pulse to pulse. We assume a Gaussian ablation crater of the form
(2)$$f\left(x\right)=1-ae^{-x^{2}}$$
where *a* is the fraction of the coating ablated with each pulse and the coating thickness is normalized to one.

Figure 8 shows the simulated ablation for a stationary sample (black, dashed) with eight consecutive pulses ablating the full coating (*a* = 1/8). The bluish curves are calculated as the sum of *n* pulses of the shape (1)
(3)$$f_{tot}\left(x\right)=1-\overset{n}{\underset{i=1}{\sum }}a\;e^{-{\left[x-x_{0}\right]}^{2}}$$
with a continuous shift of *x*_0_ after each pulse. The numbers for *n* and *x*_0_ are given in Figure 8. For a shift of *x*_0_ = 0.22 per pulse, it is not possible to ablate the full coating. The numbers shown in Figure 8 lead to the red dots in Figure 9.

Assuming this continuous shift, the results show that the setup has a shift of *x*_0_ = 0.13 or 10 μm per pulse (FWHM of ablation crater is 75 μm) at the speed of 12 m/s. At 17 m/s, the alignment is not sufficient for a reasonable depth profile.

Figure 10 shows the deviation over 15 depth profiles for a NMC622 cathode (with 5% carbon) measured at 10 m/min, each averaged over four depth profiles. The carbon content is calculated by the calibration function from Figure 5 and a depth-resolved recalibration considering the overestimation of carbon in the first pulses (Figure 6).

With no average, a high variation is seen due to inherently inhomogeneous distribution of NMC beads (see [20]). For the monitoring of a global demixing of NMC and graphite, e.g., in the drying process of the cathode, a higher average might be useful.

## 4. Discussion and Conclusions

Figure 10 shows that depth profiling on a moving sample is possible with the active retracing of the focusing optics of a LIBS setup. Alignment is crucial and limits the maximum distance for retracing. At a speed of 17 m/min, we applied 16 pulses for a depth profile with a laser repetition rate of 100 Hz. This results in a retracing distance of 45 mm plus approximate 30 mm acceleration and 30 mm deceleration. The total travel of the stage used for retracing also limits the measurable coating thickness. The deviation of a few micrometers per pulse can lead to a flat, but wider, ablation crater and can make depth profiling with many pulses or a higher speed of movement impossible. For such applications, it is highly recommended to work with high-repetition-rate lasers and spectrometers. The technique works fine for electrode foils and could also be implemented for coating analysis on galvanized steel or other fast moving industrial applications. In comparison to previous approaches, such as applying different pulse energies at consecutive positions, we think that this technique has great advantages for depth-resolved LIBS. To the knowledge of the authors, elemental analyzing techniques other than LIBS are not capable of measuring depth profiles on moving samples, so this could be a unique selling point. Further improvements could lie in a more precise alignment and a more precise tracing of the speed.

## Figures and Tables

**Figure 1 sensors-23-01082-f001:**
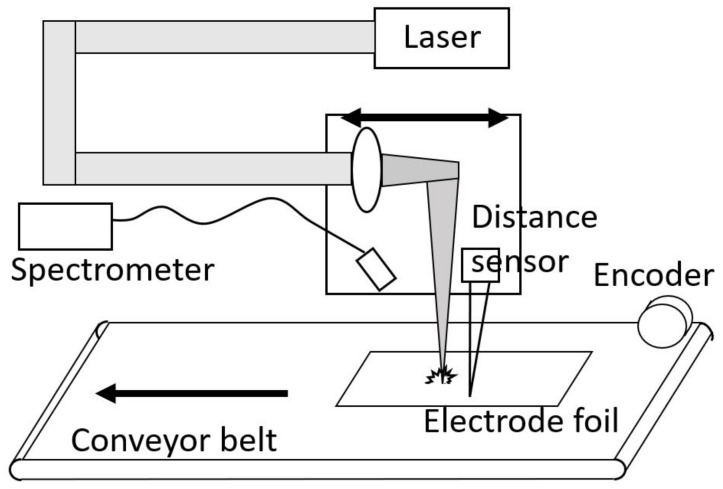
Sketch of the LIBS machine for inline depth profile measurements. The focusing lens is mounted on a motorized stage aligned along the movement of the conveyor belt. The direction of movement of the conveyor belt is indicated by the lower black arrow. The upper black double arrow indicates the movement of the LIBS setup, including the focusing lens and the deflection mirror of the pulse laser, and the detection lens for the plasma light. The synchronous movement is controlled by the signal of an encoder.

**Figure 2 sensors-23-01082-f002:**
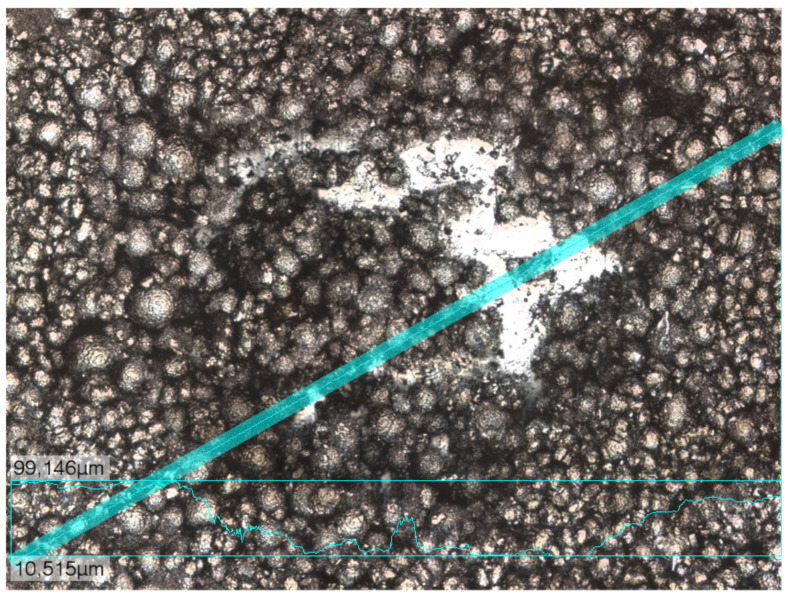
Microscope image of the nearly fully ablated coating. The shiny part shows the aluminum foil. In the central crater, the NMC beads can be seen clearly. Outside of the crater, the NMC is partly covered with graphite and binder.

**Figure 3 sensors-23-01082-f003:**
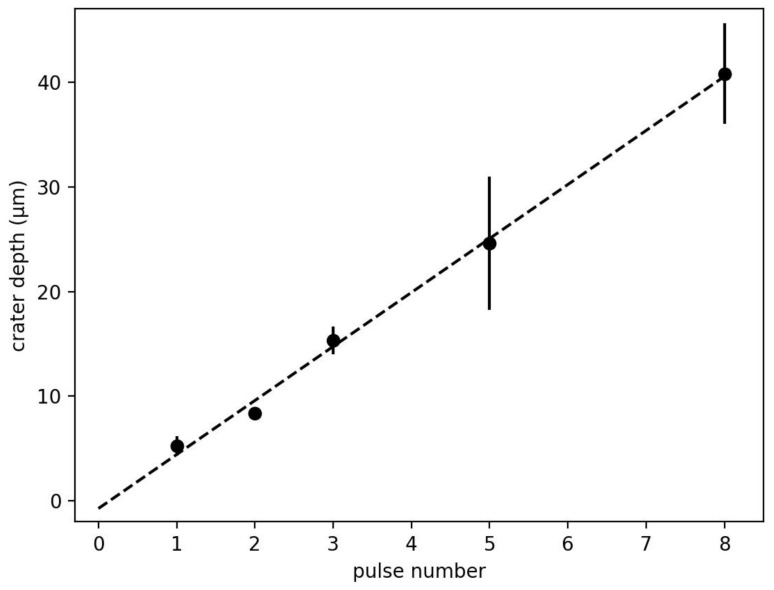
Crater depth as a function of the pulse number. Three measurements were done per pulse number; the error bar shows the standard deviation. The fit parameters are: slope 5.15 μm per pulse, offset −0.73 μm.

**Figure 4 sensors-23-01082-f004:**
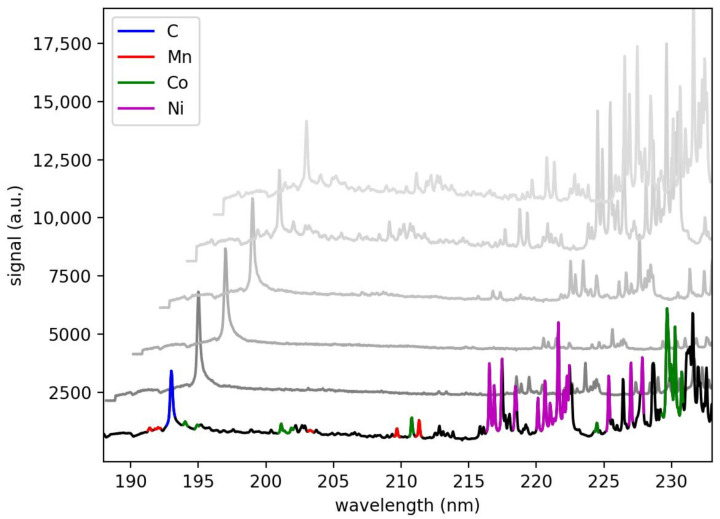
LIBS Spectrum of an NMC-cathode in the range between 190–235 nm, showing the elements Ni, Co, Mn and C averaged over 200 positions and 10 pulses per position. Colored lines are used in the evaluation. The gray shifted lines show single pulses of a depth profile (pulse 1,3,5,7,9 in rising order).

**Figure 5 sensors-23-01082-f005:**
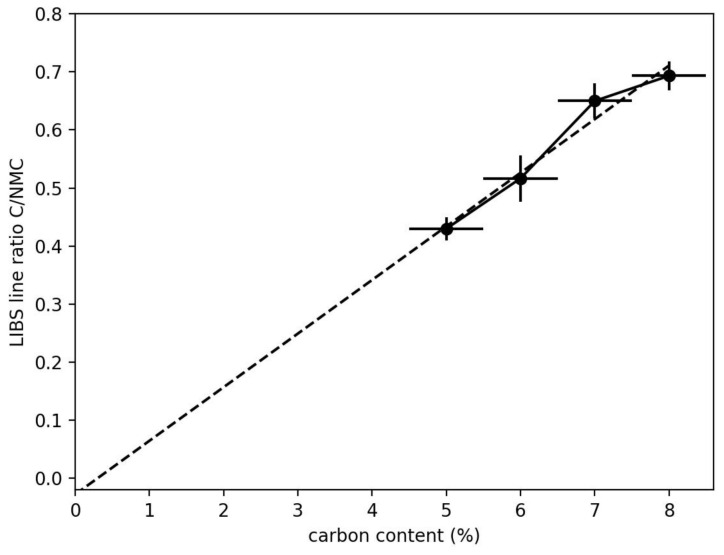
Calibration curve of the LIBS line ratio as a function of the carbon content. The error on the x-axis is the assumed error in preparation; the error on the y-axis is derived from three measurements.

**Figure 6 sensors-23-01082-f006:**
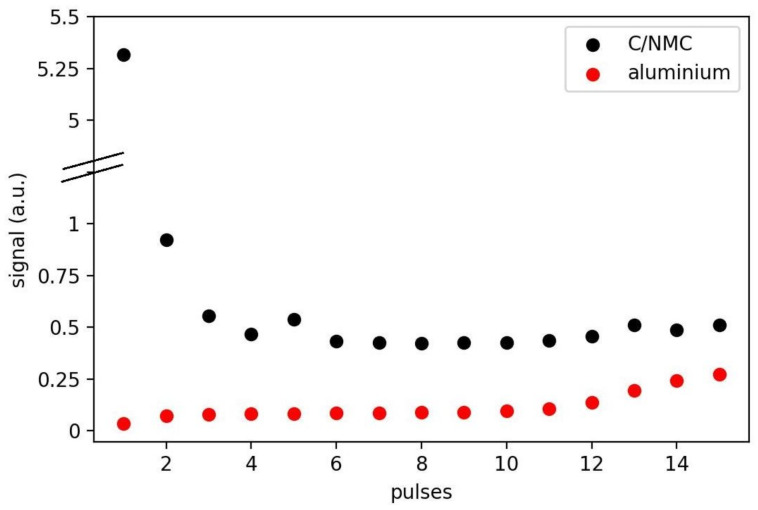
Signal of the carbon line divided by the NMC lines (black) and the aluminum signal (red) as a function of pulse number.

**Figure 7 sensors-23-01082-f007:**
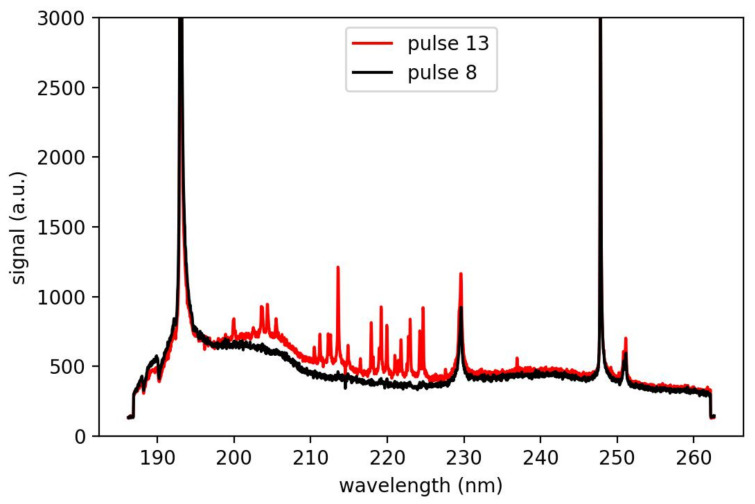
LIBS Spectrum of pulse 8 and 13 of an anode.

**Figure 8 sensors-23-01082-f008:**
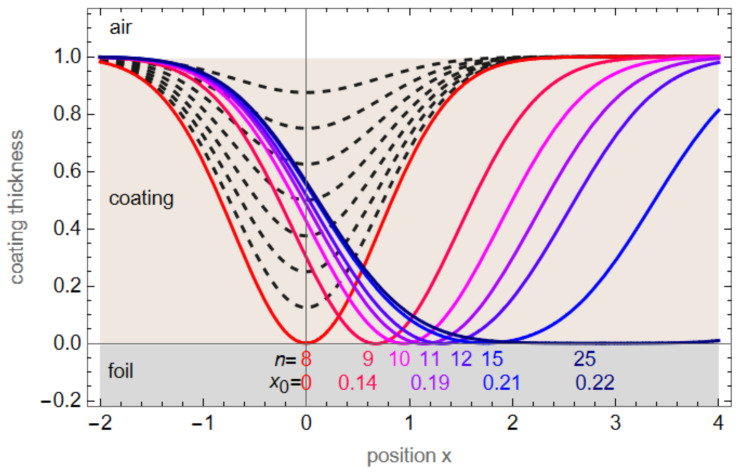
Graph of a Gaussian ablation profile. The light brown coating (y-axis from 0 to 1) is ablated with *n* = 8 consecutive pulses (dashed lines) until the crater reaches the foil. For a continuous shift *x*_0_ from pulse to pulse, more pulses are needed to ablate the coating (*n* = number of pulses).

**Figure 9 sensors-23-01082-f009:**
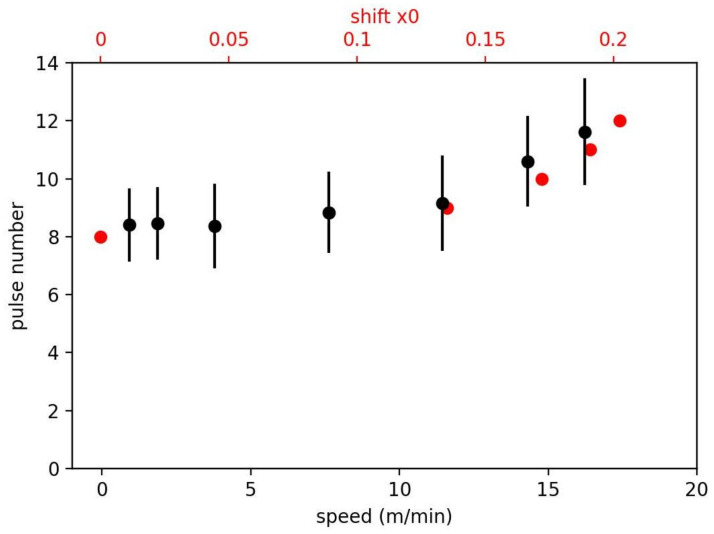
Experimental results of pulses needed to ablate the anode coating at different speeds of movement. For comparison, the red dots and the upper x-axis show corresponding simulated values according to Equation (2).

**Figure 10 sensors-23-01082-f010:**
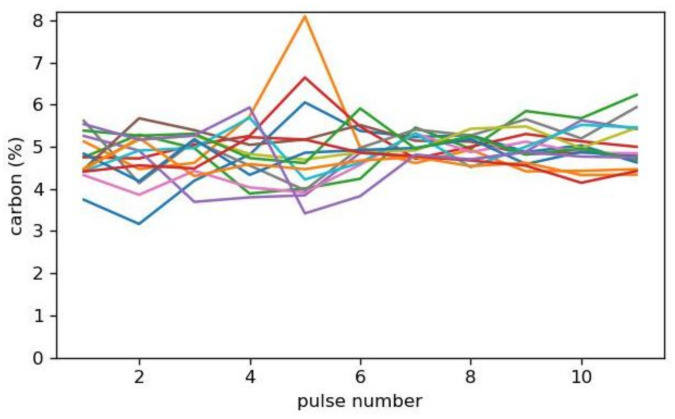
Calculated carbon content of the cathode as a function of the ablation pulse. Each graph is averaged over 4 depth profiles.

## Data Availability

Not applicable.

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
