# Peer review of "Depth-Resolved Elemental Analysis on Moving Electrode Foils with Laser-Induced Breakdown Spectroscopy"

_sensors, 2023, doi:10.3390/s23031082_

Round 1
Reviewer 1 Report
It is a original work that devoted to study of depth-resolved elemental analysis on moving electrode foils using LIBS technique. This study is correct and interesting from scientific point of view. However, there are some notes, which authors must take into consideration:
1. The use of LIBS for elemental analysis of materials and study of depth profiles is well known and described in scientific literature.
Though authors paid attention to consideration of depth-resolved elemental analysis on moving foil samples it is necessary to describe in more details the novelty of this research in comparison with other works in the field.
2. Though authors prepared sufficient list of references they missed a number of fundamental works (including pioneering ones) concerned with LIBS analysis of materials. It is necessary to complete the list of refences adding such publications.
To summarise: This research work is worth publication, but it needs some revisions.
Author Response
Thank you for carefully reading our manuscript and your suggestions to improve it.
- In the introduction we mention the differences to other research done in this field: “To our knowledge, there is no publication so far in which the depth-resolved analysis of the homogeneity of the elemental distribution in motion with LIBS has been performed.” Could you please describe more detailed what is missing?
- We have added more relevant articles concerning depth profiling on lithium-ion electrode foils and a pioneering depth profiling work from Vadillo from 1998.
Reviewer 2 Report
The authors present a study on a new method for online measurement of depth profiles on a continuously moving sample with laser-induced breakdown spectroscopy (LIBS). The results and the conclusions obtained in this study have a certain novelty. The following changes should be made before the publication of the work:
1) Authors used “laser-induced plasma spectroscopy” in the title of the manuscript. However, in all other sections, the “laser-indued breakdown spectroscopy (LIBS)” was used. It will be better to use uniform terminology throughout the manuscript.
2) As described in the section of introduction on page 2, there are some LIBS methods which are feasible for the 2D and 3D measurements of the elemental distribution in electrode foils (references 4,13–15). If this is the case, the authors should make a concise comment on the differences between those well-established methods and the method proposed here.
3) Please provide the full name of the abbreviation of “NMC” appeared firstly in the manuscript (the last paragraph in the introduction section).
4) The experimental section is much too simple. Many essential details are missing. For example, the authors should provide the information of laser parameters such as the spot size on sample surface, the laser beam profiles (as a basis for the following assumption of a Gaussian ablation crater of the Eq (1) form), and the distance from focal plane to sample surface.
5) Is there any special reason for that the integration time of the spectrometer used in this study starts before the laser pulse? It is well known that, in a standard LIBS experiment, the integration time of the spectrometer starts after the laser pulse (To achieve an optimum signal-to-noise ratio, a time delay is usually required in the range of hundreds to thousands of nanoseconds.)
6) The spectrum shown in Figure 3 represents an accumulated spectrum by many single shots? If this is the case, the number of shots should be mentioned.
7) Why error bars of the measured ratio of line intensity between carbon line and metal lines are not included Figure 4 and Figure 5?
8) It is not clear which specific metal lines have been used to derive the ratio data of line intensity plotted in Figure 4 and Figure 5 and how to extract the intensities of the carbon line and the metal lines from the measured LIBS spectra.
The work should be of interest to readers concerned with the depth-resolved elemental analysis on moving samples using LIBS. I recommend the paper for publication after minor revisions.
Author Response
Thank you for carefully reading our manuscript and your suggestions to improve it. We tried to impement all suggestions:
- Plasma changed to Breakdown.
- We have added another sentence to the difference in the introduction.
- It was explained in the next sentence. It is corrected now.
- We have added spot size and focal length of the focussing lens.
- The signal to noise is most important if you want to measure low concentration (ppm). We have high concentration (x %). By starting the spectrometer before the laser we do not need to care for the precise timing.
- Number of averaged spectra is now mentioned.
- We have added errors bars and described the error in the caption of the new figure 5.
- The evaluation is now described in chapter 3.1
Reviewer 3 Report
The manuscript describes a new method for the inline measurement of depth profiles on a continuously moving sample with laser-induced breakdown spectroscopy. The experimental results are interesting. However, the mechanisms are still unclear to me after reading the paper. I would suggest the authors consider the following points:
1. In the part on Materials and Methods, the author mentioned that the nanosecond laser with 1064 nm and 6 ns pulse duration, 10 mJ energy, and 100 Hz was used. Are these parameters chosen for a reason? Are these factors decisive?
2. In Figure 2, the author thinks that the Crater depth and the pulse number are linear. According to this statement, 0 pulses should correspond to a thickness of Zero, which is obviously not the case in the figure. Should authors refit?
3. When the authors discuss the results of Figure 3, they mention 265 nm, 249 nm, 260 nm, etc., but Figure 3 only shows about 187 nm to 233 nm. How are these conclusions reached?
4. As stated by the authors, this measurement method is of great significance for the production of electrode foils for lithium-ion batteries to monitor the homogeneity of its constituents. Can this technology be applied to other materials in similar fields?
Author Response
Thank you for carefully reading our manuscript and your suggestions to improve it.
- The laser parameters are tuned in a way to ablate the coating fast, with only a few pulses but still some depth resolution. This is the case at 10mJ. We added a sentence in the beginning of chapter 3.1. The discussion in chapter 3.2 and chapter 4 explain that the total measurement time is crucial for the alignment. A higher repetition rate is preferable, but 100 Hz is the fastest we have.
- The offset is -0.73µm, which is less than the shown error bars of the measurements. We have added the fit parameters in the caption.
- We have changed the text, according to the used lines and added a sentence to the comparison to NIST.
- In the discussion we state that “The technique works fine for electrode foils and could also be implemented for coating analysis on galvanized steel or other fast moving industrial applications.”
Author Response
Thank you for carefully reading our manuscript and your suggestions to improve it. We tried to impement all suggestions:
- A more detailed description is made in the caption of figure 1.
- We have included an image of the ablation crater.
- LIBS spectra of different layers are included in figure 4 now.
- A better description is included in chapter 2
- The model in figure 7 is compared to experimental data in figure 8 (now figure 9)
Reviewer 5 Report
In this paper, the authors focused on the depth-resolved elemental analysis on moving electrode foils with LIBS. The authors have done some detailed and significant researches. Especially, the authors compared the ablation coating thickness by theoretical with experimental data. The theoretical results shown a good agreement with experimental results. The work should be refereed for the depth analysis of targets by LIBS. The writing of the manuscript is also good.
Many of the LIBS work have been performed on the depth analysis of the targets. However, they were not mentioned in the manuscripts, such as Journal of Cultural Heritage 58 (2022) 237–244, J. Anal. At. Spectrom., 2019, 34, 226, Acta Photonica Sinica,2021,50(10):1030003 and so on. Thus, the author should be referred and citied some works of them to introduced the progress of the depth analysis by LIBS in introduction.
In sum, I recommend to accepted it after the necessary revision.
Author Response
Thank you for carefully reading our manuscript and your suggestions to improve it.
We have added to more relevant articles concerning depth profiling on lithium-ion electrode foils and a pioneering depth profiling work from 1998. LIBS in the context of cultural heritage is important and worth mentioning but not the focus of this work.
Round 2
Reviewer 4 Report
This manuscript has included the things that I had asked for and answered the questions that were asked.